# Species abundance information improves sequence taxonomy classification accuracy

Benjamin D. Kaehler [1,2,8]*, Nicholas A. Bokulich [3,4,8]*, Daniel McDonald[5], Rob Knight [5,6,7], J. Gregory Caporaso [3,4]* & Gavin A. Huttley [1]*

Popular naive Bayes taxonomic classifiers for amplicon sequences assume that all species in the reference database are equally likely to be observed. We demonstrate that classification accuracy degrades linearly with the degree to which that assumption is violated, and in practice it is always violated. By incorporating environment-specific taxonomic abundance information, we demonstrate a significant increase in the species-level classification accuracy across common sample types. At the species level, overall average error rates decline from 25% to 14%, which is favourably comparable to the error rates that existing classifiers achieve at the genus level (16%). Our findings indicate that for most practical purposes, the assumption that reference species are equally likely to be observed is untenable. q2-clawback provides a straightforward alternative for samples from common environments.

[1] Research School of Biology, Australian National University, Canberra, Australia. [2] School of Science, University of New South Wales, Canberra, Australia. [3] Center for Applied Microbiome Science, The Pathogen and Microbiome Institute, Northern Arizona University, Flagstaff, AZ, USA. [4] Department of Biological Sciences, Northern Arizona University, Flagstaff, AZ, USA. [5] Department of Pediatrics, University of California San Diego, La Jolla, CA, USA. [6] Department of Computer Science and Engineering, University of California San Diego, La Jolla, CA, USA. [7] Center for Microbiome Innovation, University of California San Diego, La Jolla, CA, USA. [8] These authors contributed equally: Benjamin D. Kaehler, Nicholas A. Bokulich. *email: b.kaehler@adfa.edu.au; nicholas.bokulich@nau.edu; gregcaporaso@gmail.com; gavin.huttley@anu.edu.au

Advances in high-throughput DNA sequencing and bioinformatics analyses have illuminated the crucial roles of microbial communities in human populations and planetary health[1,2] and enable microbiome meta-analysis on a massive scale[3]. An important step in characterizing microbial communities is classification of short marker-gene DNA sequences (e.g., bacterial 16S rRNA genes) to infer taxonomic composition.

Short marker-gene sequence reads often contain insufficient information to differentiate species using conventional methods[4–8]. However, current best practices rely on species-level classification to circumvent well-documented inconsistencies between genus-level reference taxonomies and molecular phylogeny (e.g. *Clostridium* and *Eubacterium*)[9].

In this work, we demonstrate that a substantial improvement in classification accuracy of marker-gene sequences can be achieved if a reference taxonomic distribution for the sample's source environment is known. This technique enables marker-gene sequencing to differentiate individual species at a level of accuracy previously only available at the genus level.

We focus on q2-feature-classifier, a QIIME 2[10] plugin for taxonomic classification. In previous work[4] we benchmarked this method against other common classifiers, including the RDP Classifier[11] and several consensus-based methods using real and simulated data for four bacterial and fungal loci. We also tested a developmental feature that showed that knowing the mixing proportions of mock communities improved taxonomic classification accuracy. In general, q2-feature-classifier meets or exceeds the accuracy of the other classifiers[4]. However, all tested methods perform similarly if their parameters are tuned in a concordant manner. Significant performance enhancement demonstrated in the current work for q2-feature-classifier therefore imply improved performance over those other methods.

## Results

**Taxonomic weight assembly with q2-clawback**. The RDP Classifier and q2-feature-classifier use similar naive Bayes machine-learning classifiers to assign taxonomies based on sequence k-mer frequencies, and exhibit very similar performance when default parameters are used[4]. The default assumption of these classifiers is that each species in the reference taxonomy is equally likely to be observed. Unlike the RDP classifier, however, q2-feature-classifier now allows prior probabilities to be set for each species. We refer to the prior probabilities as taxonomic weights and the default equal probabilities as uniform weights. We hypothesized that inputting the frequencies with which each taxon is actually observed in nature as taxonomic weights would improve classifier performance.

Taxonomic weights were downloaded and assembled using our new utility, q2-clawback (https://github.com/BenKaehler/q2-clawback). We created weights for 14 Earth Microbiome Project Ontology (EMPO) 3 habitat types[1] across 21,513 samples from the Qiita microbial study management platform[3] (see Methods for details). q2-clawback can assemble weights from any appropriately curated set of samples or by querying Qiita on any available metadata category. We refer to EMPO 3 habitat-specific taxonomic weights as bespoke weights.

**Taxonomic weights improve species classification**. To test classification accuracy using varying taxonomic weights, we developed a novel cross-validation strategy that accounted for the observed abundances of taxa in any given habitat. This strategy ensured that a classifier was never asked to classify a sequence that had occurred in its training set or generate taxonomic abundances that had directly contributed to its input taxonomic weights. To our knowledge, our cross-validation strategy is the first to incorporate information about taxonomic weights in assessing taxonomic classifier performance. This situation is known in machine learning as imbalanced learning[12]. See Methods for a thorough description of the test dataset and cross-validation procedure.

Bespoke weights achieved significantly better species-level classification accuracy than other taxonomic weight strategies. Bespoke weights significantly outperformed uniform weights when both were compared at the species level across the 14 EMPO 3 habitats (bespoke error rate = 14%, uniform error rate = 25%, paired $t$-test $P = 5.8 \times 10^{-5}$) (Fig. 1). Similar results were obtained for Bray-Curtis dissimilarity and F-measure (see Supplementary Notes). Averaged across the 14 EMPO 3 habitats, Proteobacteria and Firmicutes were the most abundant phyla (34% and 18% of reads, respectively). Switching from uniform to bespoke weights caused error rates for classification of species in these phyla to drop from 35.4% (±0.7% s.e.) to 22.3% (±0.4% s.e.) and 43.6% (±0.7% s.e.) to 24.3% (±0.3% s.e.), respectively (Supplementary Fig. 5). These differences were highly significant for both Proteobacteria and Firmicutes (paired $t$-tests $P = 1.4 \times 10^{-6}$ and $P = 8.4 \times 10^{-6}$, respectively).

Using bespoke weights, researchers can now classify sequences at the species level with the same confidence that they previously classified sequences at the genus level (Fig. 1). To demonstrate that bespoke classification achieves both greater accuracy and greater depth of taxonomic classification, we compared the error rates of bespoke classification at the species level to uniform classification at the genus level. The mean error rate (the proportion of reads incorrectly classified) across the 14 EMPO 3 habitat types was 14% (±1% s.e.) for bespoke weights at the species level and 16% (±1% s.e.) for uniform weights at the genus level. These results indicate that bespoke weights achieve comparable or better species-level accuracy to what uniform weights can only accomplish at the genus level. (As described below, bespoke weights significantly outperform uniform weights by all metrics when both are compared at the species level.) The mean Bray-Curtis dissimilarity between observed and expected taxonomic abundances was 0.13 (±0.01 s.e.) for bespoke weights at the species level and 0.15 (±0.01 s.e.) for uniform weights at the genus level (single-sided paired $t$-test $P = 0.013$) (Supplementary Table 2, Supplementary Fig. 2), indicating better performance of bespoke weights. See Supplementary Notes for more details of our benchmarking results.

**Bespoke accuracy boost is correlated to weight fitness**. In addition to testing classification accuracy using cross validation within EMPO3 habitats, we performed a set of additional experiments to evaluate how the selection of taxonomic weights impacts classifier performance. First, we assessed classification accuracy when using the average of the 14 EMPO 3 habitat-specific bespoke weights, which we term average weights. For every EMPO 3 habitat, bespoke weights outperformed average weights (sign test $P = 6.1 \times 10^{-5}$) (Fig. 2, Supplementary Figs. 2 and 3). Similarly, average weights always outperformed uniform weights (sign test $P = 6.1 \times 10^{-5}$) (Fig. 2, Supplementary Figs. 2 and 3). The implication is that classification accuracy improves when taxonomic weights more closely resemble taxonomic frequencies observed in nature. Second, we assessed classification accuracy when training classifiers with taxonomic weights from the EMPO 3 habitats other than the sample's source habitat, which we term cross-habitat weights. Importantly, uniform weights demonstrated inferior performance to cross-habitat weights in 117 out of 182 cross-habitat comparisons, suggesting that any type of naturally derived taxonomic weight has the

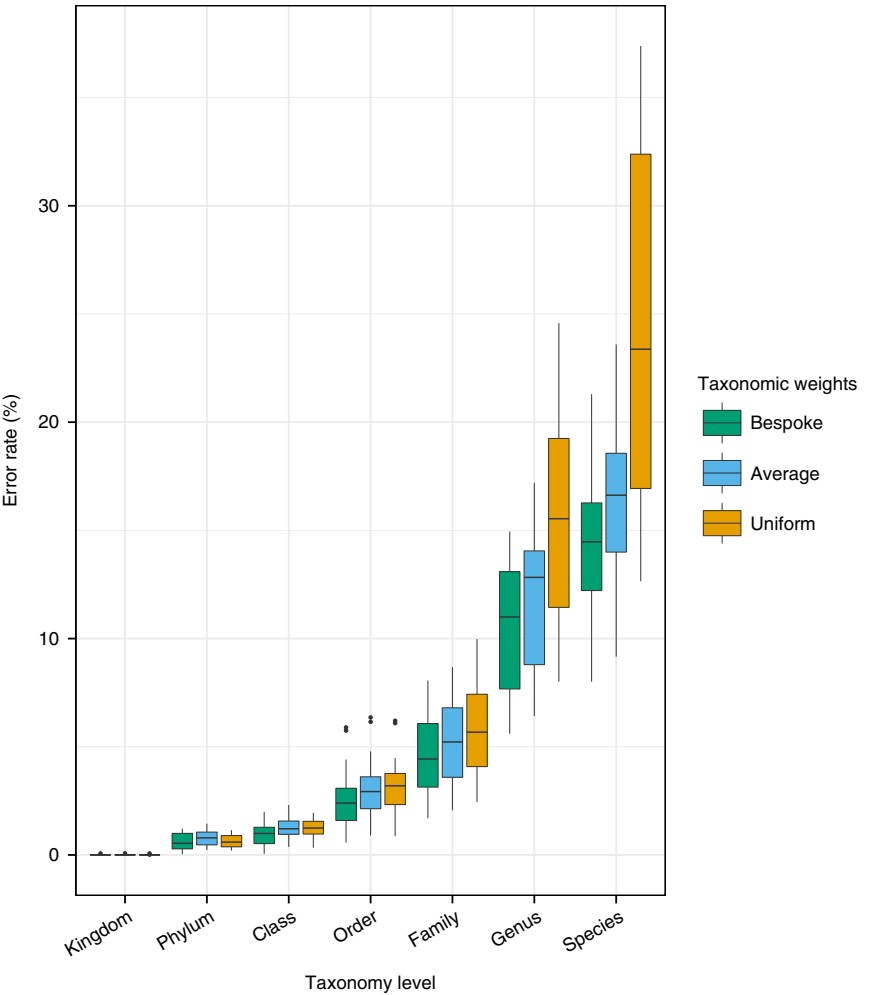

**Fig. 1** Habitat-specific taxonomic weights classify sequences at the species level with the same confidence that uniform weights classify at the genus level (single-sided paired t-test, species bespoke vs genus uniform, $t = 1.6$, $P = 0.14$). Overlaid columns show average proportion of incorrectly classified reads for three taxonomic weighting strategies at each taxonomic level. Bespoke weights were habitat-specific. Average weights were averaged across 14 EMPO 3 habitats. Uniform weights are the current best practice. Box plots are across 14 EMPO 3 habitats. Box bounds and centre lines show first and third quartiles and median. Whiskers extend to measurements no further than 1.5 times the interquartile range from the nearest quartiles. Outliers are plotted individually. 21,513 empirical taxonomic abundances contributed to the results. Source data are provided as a Source Data file

potential to improve classification accuracy, even if those weights were derived from a dissimilar habitat type (Fig. 3; see Supplementary Notes). The degree to which cross-habitat taxonomic weights could accurately classify the taxa within a given environment was proportional to the similarity between those weights and the bespoke weights for the source habitat: as the taxonomic weights moved away from the bespoke weights for a given sample, error rate increased (Pearson $r^2 = 0.57$, $P < 2.2 \times 10^{-16}$) (Fig. 4; see Supplementary Notes and Methods).

The ability of uniform-weight classifiers to resolve species-level differences from marker genes is directly related to sequence similarity among the reference species. Species with highly similar sequences will be difficult to differentiate, even if these species occupy exclusive ecological habitats. However, bespoke weights incorporate habitat-specific species distribution information to guide sequence classification. Hence, classification accuracy under bespoke weights for a given habitat type is tied to sequence similarity and the distribution of individual species in that habitat. We devised a statistic that we term the confusion index to quantify how often similar sequences originated from different species in the same habitat (see Methods). The confusion index is a function of the taxonomic difference between sequences with

similar k-mer profiles and the frequency that they appear, taking the bespoke weights as the likelihood of observing a given species. We found that error rates for bespoke weights were correlated with the confusion index (Fig. 5; Pearson $r^2 = 0.72$, $P = 1.4 \times 10^{-4}$, see Methods and Supplementary Notes). That is, classification accuracy is affected by how often different species in the same sample have similar amplicon sequences but different taxonomic classifications, and that varies between EMPO 3 habitats.

The clear logic behind using bespoke weights is to encourage the classifier, when faced with uncertainty, to err on the side of taxa that are more abundant for a given habitat. The risk with this approach is that less abundant taxa may then be neglected. We tested this possibility by using the qualitative performance metrics taxon detection rate (TDR) and Taxon Accuracy Rate (TAR)[4] which take only presence and absence data into account. For TAR, bespoke weights outperformed average weights and average weights outperformed uniform weights on average (Supplementary Fig. 7). The differences were not significant (minimum paired t-test $P = 0.17$). For TDR, the trend fit with all of our other tests with average weights always outperforming uniform weights and bespoke weights always outperforming average weights (sign test $P = 6.1 \times 10^{-5}$) (Supplementary Fig. 8). While the TAR

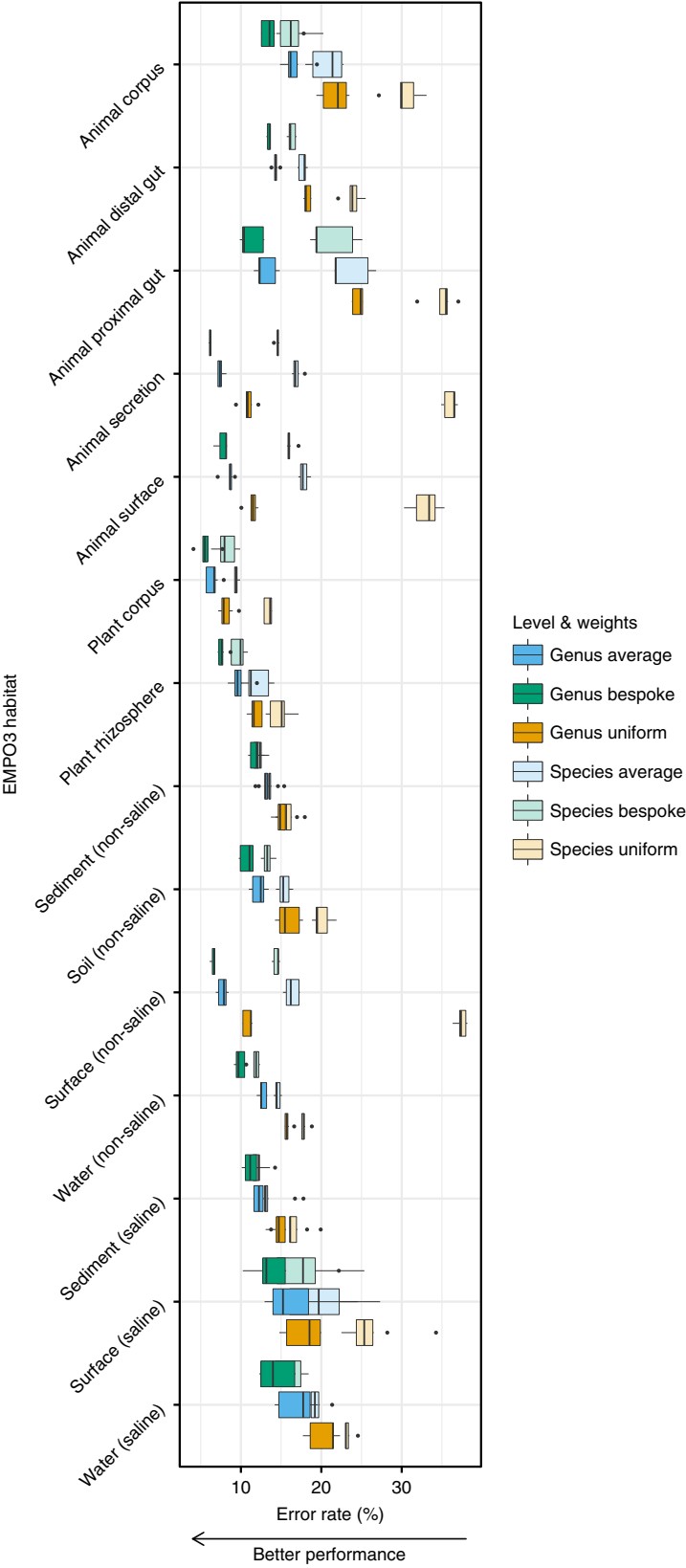

**Fig. 2** Bespoke weights outperform average weights across EMPO 3 habitat types, and average weights outperform uniform weights (sign test $P = 6.1 \times 10^{-5}$). Columns show average proportion of incorrectly classified reads for differing taxonomic weighting strategies and at genus and the species levels. Bespoke weights were habitat-specific. Average weights were averaged across the 14 EMPO 3 habitats. Uniform weights are the current best practice. Tests were based on 5-fold cross validation across 18,222 empirical taxonomic abundances. Box plots are across cross-validation folds. Box bounds and centre lines show first and third quartiles and median. Whiskers extend to measurements no further than 1.5 times the interquartile range from the nearest quartiles. Outliers are plotted individually. Source data are provided as a Source Data file

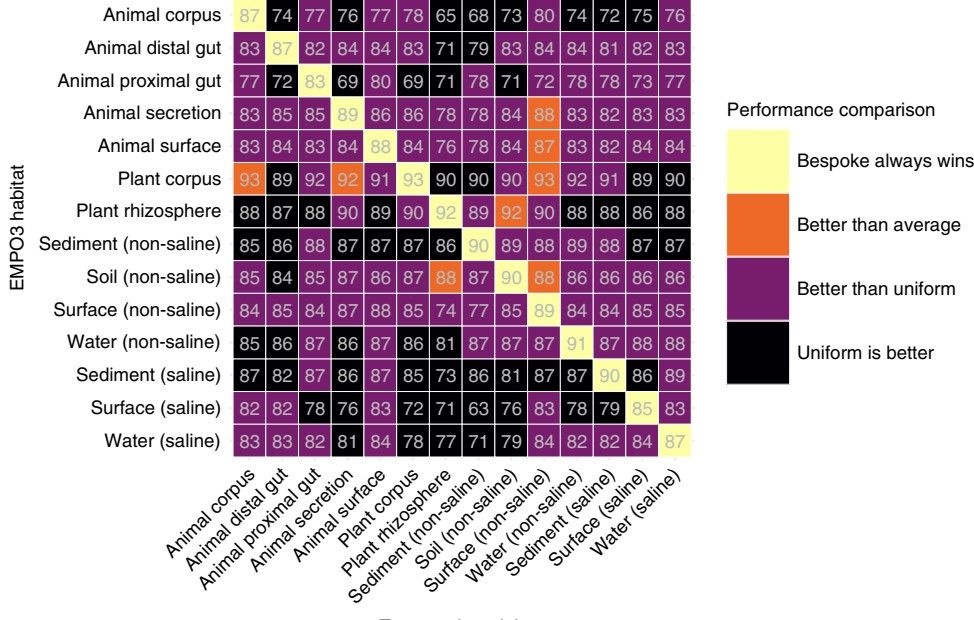

**Fig. 3** Summary of the effect of using cross-habitat weights (taxonomic weights different to the sample habitat). Light grey numbers show F-measure as a percentage. Bespoke weights (when taxonomic weights match sample habitat) are always superior. Occasionally (8 times) cross-habitat weights achieved higher accuracy than weights that were averaged across the 14 EMPO 3 habitats. Frequently the cross-habitat weights were better than uniform weights (which is current best practice, 109 times). Occasionally the uniform weights outperformed the cross-habitat weights (65 times). Source data are provided as a Source Data file

results were not as positive as our other results, they nonetheless show that no penalty was incurred, even judging purely by presence and absence of taxa, when using bespoke weights.

We also tested whether using bespoke weights would reduce classification accuracy for rare taxa. By averaging results across the 14 EMPO 3 habitats, we plotted average differences in uniform and bespoke weight error rates against average species abundance (Supplementary Fig. 9). We found that for abundances less than $10^{-4}$, there was some evidence of degradation in classifier accuracy, although for most species it made no difference. For a typical sample depth in the order of $10^4$ reads, this indicates that there may be some risk of misclassification for singletons.

## Discussion

The assumption of uniform weights, that species are evenly distributed in nature and hence equally likely to be detected, is incorrect. We have demonstrated that this assumption imposes a consistently negative impact on performance, even when compared to deliberately incorrect taxonomic weights selected from ecologically dissimilar environmental sources (the cross-habitat weights). As a result, we suggest the continued usage of uniform weights is not justifiable. When publicly accessible pre-existing microbiome data is available for the sample (i.e., environment) type being investigated, bespoke weights should be used. For other natural sample types that lack sufficient characterization for bespoke weight assembly, average weights estimated from global microbial species distributions are superior to uniform weights. We did observe some degradation in performance for rare species. If the presence or absence of singletons for a typical sample is critical to experimental design, then we advocate using amplicon sequence variants[13,14] rather than taxonomic classification. For highly unusual sample distributions, e.g., in synthetic populations, we recommend compiling custom bespoke weights from existing samples. We demonstrated species-level classification improvement with as few as

**Table 1 Sample counts for each EMPO 3 habitat type**

| EMPO 3 habitat type | Number of samples |
| --- | --- |
| Animal corpus | 1158 |
| Animal distal gut | 5632 |
| Animal proximal gut | 903 |
| Animal secretion | 974 |
| Animal surface | 1,839 |
| Plant corpus | 543 |
| Plant rhizosphere | 328 |
| Sediment (non-saline) | 188 |
| Surface (non-saline) | 1383 |
| Soil (non-saline) | 2802 |
| Water (non-saline) | 4769 |
| Sediment (saline) | 414 |
| Surface (saline) | 152 |
| Water (saline) | 428 |

122 samples (four-fifths of the saline surface samples from the EMP study; Table 1).

Our key finding is that taxonomic classification is sensitive to taxonomic weight assumptions, and better alternatives to assuming uniform weights exist for natural samples. Even when we intentionally used taxonomic weights from the wrong habitat type (our cross-habitat tests), these weights still outperformed uniform weights for species classification in the majority of cases. Where uncertainty exists regarding the correct choice of habitat for taxonomic weights, average weights offer a generalized solution for improved accuracy over uniform weights. Systematic comparison of uniform, average, bespoke, and cross-habitat weights demonstrated that the more specific the taxonomic weights are to a query sample's environment, the better the classification accuracy. Thus, taxonomic weight selection impacts classification results, but any deliberate decision regarding choice of taxonomic weights is unlikely to negatively impact classification accuracy beyond the penalty imposed by uniform weights.

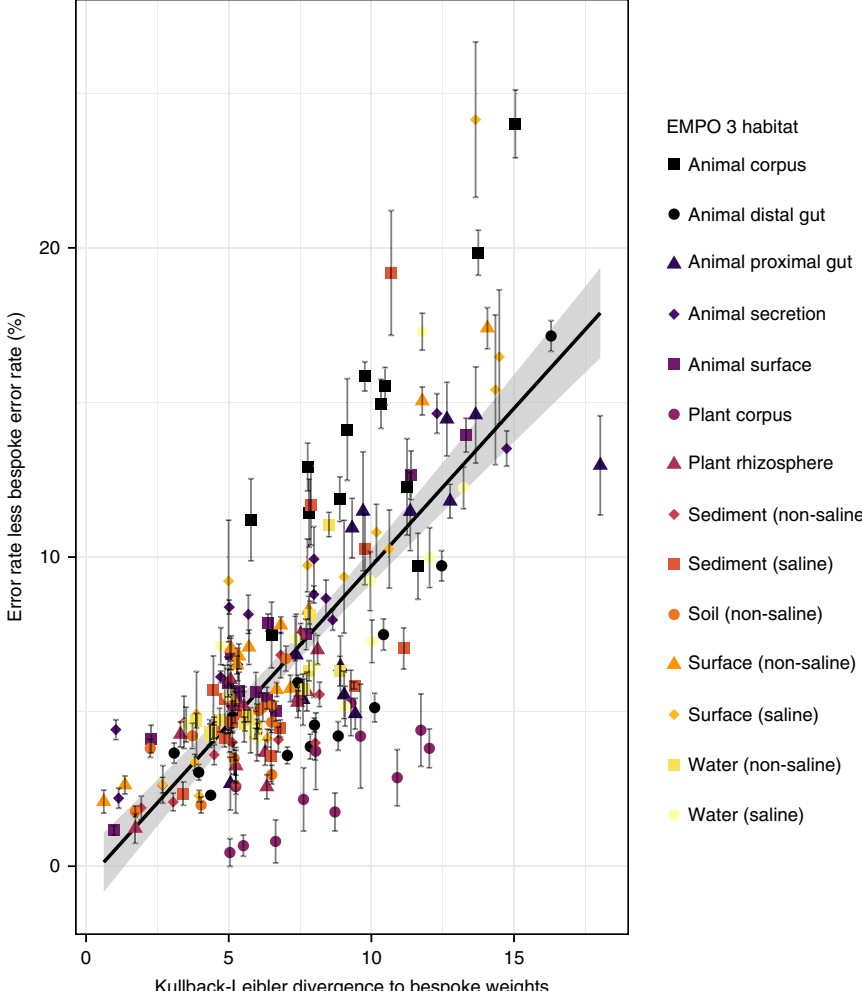

**Fig. 4** Classification accuracy degrades as taxonomic weights diverge from sample abundances. Cross testing of classification accuracy by setting taxonomic weights to those from each of the 13 EMPO 3 habitats other than the appropriate bespoke weights, across 14 EMPO 3 habitats. There is a clear association (Pearson $r^2 = 0.57$, $P < 2.2 \times 10^{-16}$). Error bars show standard error. Source data are provided as a Source Data file

q2-clawback facilitates the use of appropriate weights by making it easier for the researcher to assemble weights that are specific to a particular sample type, provided that appropriate source data are available. For instance, it is trivial to assemble weights for all stool samples with human hosts from Qiita (See the online tutorial, https://library.qiime2.org/plugins/q2-clawback).

In common with other methods, bespoke classification is not immune to errors that result from poorly curated reference data (e.g., reference sequence misannotation). The use of empirical species distributions also creates a potential source of error for bespoke classification (e.g., misannotation of sample types or sequencing biases in empirical samples could be propagated if those samples were used to develop taxonomic weights). Constructing taxonomic weights via meta-analysis of many studies of a single environment type, as we perform here and democratize with q2-clawback, reduces the impact of unsystematic errors such as mislabelled samples. More systematic errors, for instance from bias present in common sequencing techniques, could be controlled by integrating multiple technologies for microbial distribution estimation (e.g., marker gene, shotgun metagenome, metaproteome, and non-molecular methods). In the Supplementary Information, we demonstrate that shotgun metagenome data may be used to construct bespoke taxonomic weights (Supplementary Fig. 6). Further work could provide an interesting route for inferring more realistic approximations of natural taxonomic weights.

Regardless of the source of classification error, as bespoke classification typically starts from the raw reference and read data when weights are derived, its use does not lead to classification errors being propagated through history. Efforts to curate reference databases and the continued contribution of researchers to online microbiome data repositories will help refine and extend our ability to apply appropriate bespoke weights for sequence classification in diverse sample types. The magnitude of improvement in classification accuracy and robustness to the use of deliberately inappropriate weights in our cross-habitat tests make us confident that these sources of error are of secondary importance to the much larger error of assuming uniform weights.

The results we present provide a general path for delivering species-level classification accuracy. As such, the work provides a complementary solution to the small number of existing specialist classification databases[15–18]. Moreover, bespoke weight classification permits the detection of unexpected species not encompassed by custom databases.

By improving species-level classification of marker-gene sequences, bespoke weights may support critical functional inferences, e.g., differentiation of pathogenic and non-pathogenic species of the same genus[19–24]. Ongoing improvements in public

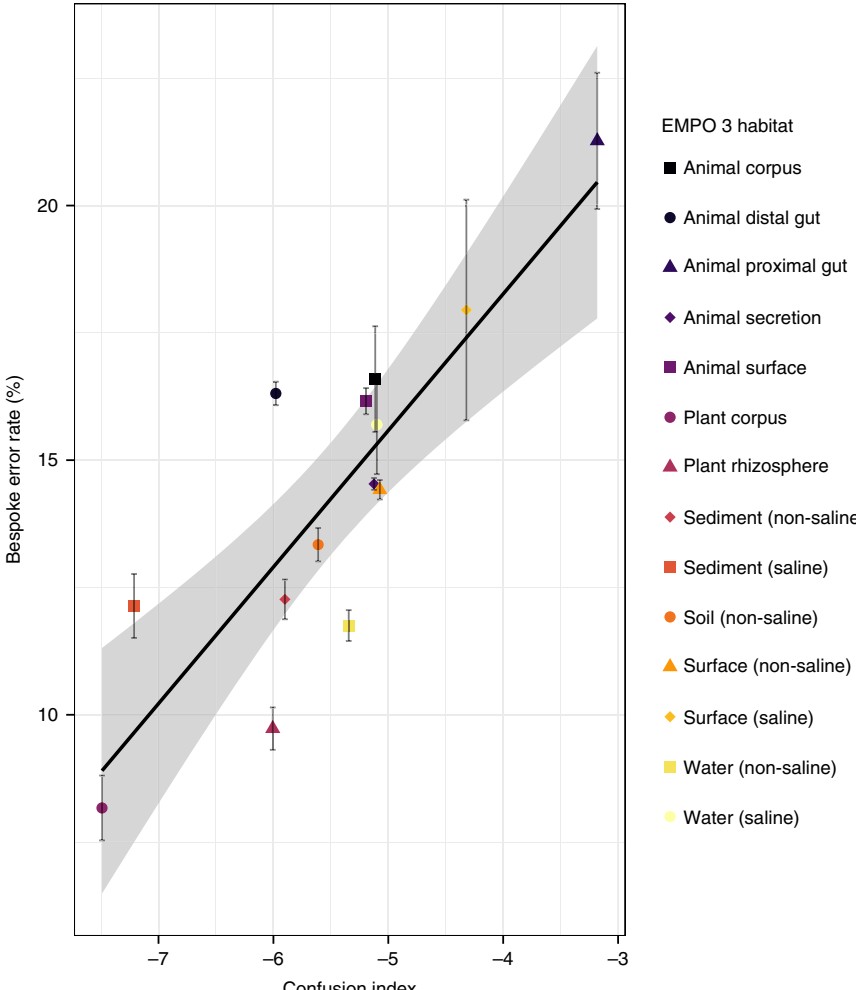

**Fig. 5** Classification accuracy when using the appropriate bespoke weights is largely explained by how often sequences from different from species are confused (Pearson $r^2 = 0.72$, $P = 1.3 \times 10^{-4}$). The confusion index is the log of the expected level of taxonomic difference between two similar reference sequences weighted by the likelihood of observing similar sequences. All points calculated using 5-fold cross validation. Error bars are standard errors across folds. Regression confidence intervals are 95%. Source data are provided as a Source Data file

reference sequence and sample databases will further boost performance, supporting biological insight into global microbiome compositions. Uniform weights should always be avoided, as they distort natural species distributions, leading to imprecise and incorrect taxonomic predictions.

## Methods
**Data**. We downloaded all public 150 nucleotide 16S v4 samples for 18 EMPO 3 habitat types from Qiita[3] using q2-clawback. The downloaded data consisted of sequence variant and abundance information. The sequence variants were prepared by the standard Qiita pipeline, including Deblur[13], prior to download. q2-clawback uses redbiom[25] (https://github.com/biocore/redbiom) to access Qiita. Data from the following Qiita studies were used: 11113[26], 11444, 1716, 10369[27], 990[28], 2080, 1713, 894, 1289, 1883, 1673, 1288, 10353, 2192[29], 10323, 678, 1773, 662, 1799, 864, 1481, 1024[30], 1064, 2182, 10934, 1674, 1795[31], 10273, 10283[32], 10422[33], 804, 10308, 1056[34], 2382[30], 1240, 889, 1041, 1717, 1222, 11149, 11669, 807[35], 10245, 1711, 1721, 910, 1001, 895, 550[36], 1747[37], 713[38], 755, 861, 958[39], 11161[40], 11154[41], 945, 723, 1715, 1714, 10798.

The three EMPO 1 control EMPO 3 habitat types were excluded, as well as Hypersaline (saline), Aerosol (non-saline), and Plant surface, which all had fewer than nine samples in the Qiita database for 150 nt sequence variants. The number of samples downloaded for each EMPO 3 habitat is shown in Table 1.

For the cross validation analysis, sequence-variant level data was discarded and only taxonomic abundance information was retained. The sequence variants were classified using the standard q2-feature-classifier naive Bayes classifier based on Greengenes 99% identity OTU reference data[42] to obtain empirical taxonomic abundance data for each sample. The naive Bayes classifier was trained using the balanced parameter recommendations given in Bokulich, Kaehler et al.[4].

For the shotgun data experiment (see Supplementary Notes), data was downloaded from the Human Microbiome Project website[2]. The downloaded tables had been prepared using a pipeline leading to MetaPhlAn2[43]. Paired 16S stool samples were downloaded from Qiita[3] in the form of DNA sequencing data with quality scores. The 16S samples were trimmed to 340 nt and denoised using DADA2[14]. In total, 71 pairs of shotgun and 16S stool samples were found. Reference data sets were downloaded from the NCBI RefSeq database[44]. Full 16S sequences were trimmed to the V3-V5 regions (forward primer CCTACGGGAGGCAGCAG; reverse primer CCGTCAATTCMTTTRAGT), using q2-feature-classifier[4], resulting in 20,696 reference sequences across 14,777 taxa. It should be noted that this experiment is intended for demonstration only, and that we are not advocating the use of the NCBI 16S RefSeq database for this purpose, as on average there are less than two reference sequence examples for each taxon.

**Clawback**. q2-clawback is a free, open-source, BSD-licensed package that is available on GitHub (https://github.com/BenKaehler/q2-clawback). It includes methods for downloading sequence variants from Qiita (fetch-Qiita-samples), extracting sequence variants for taxonomic classification (sequence-variants-from-samples), and assembling taxonomic weights from collections of samples of taxonomic abundance (generate-class-weights). These methods can be run independently or combined into a single method call (assemble-weights-from-Qiita). Figure 6 shows the workflow for these methods. An online tutorial is available (https://library.qiime2.org/plugins/q2-clawback).

In general, taxonomic weights are assembled as follows. A set of sequence variants with abundances are acquired (fetch-Qiita-samples). The sequence variants are extracted (sequence-variants-from-samples) and classified using the naive Bayes classifier under uniform weights using balanced settings[4]. Classification to species level is forced by setting the confidence parameter to −1. The resulting read counts are aggregated, normalised, and added to a small ($10^{-6}$

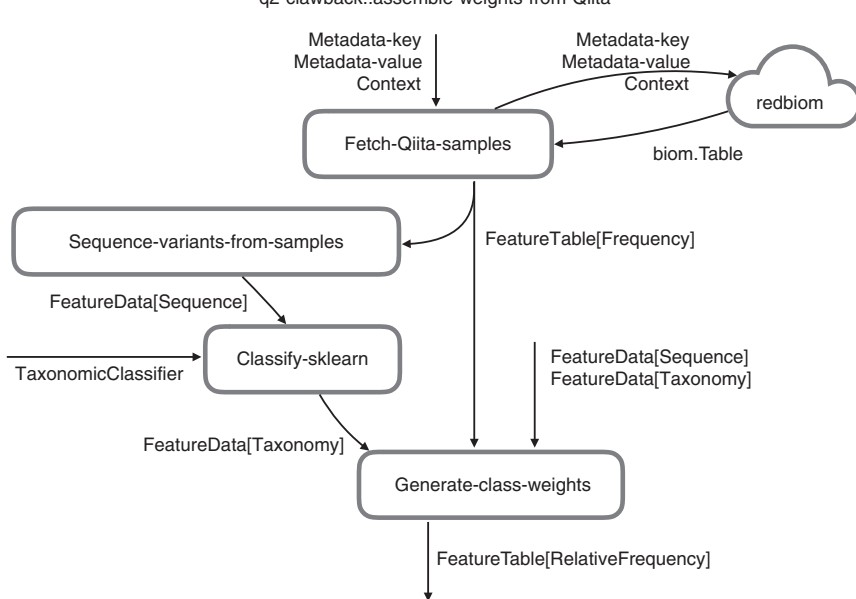

**Fig. 6** Relationship between q2-clawback methods. q2-clawback contains methods for downloading and assembling taxonomic weights. The assemble-weights-from-Qiita method wraps the illustrated workflow, but fetch-Qiita-samples, sequence-variants-from-samples, and generate-class-weights can also be accessed directly. Labelled data flows show QIIME 2 semantic types and parameters. classify-sklearn is provided by the q2-feature-classifier plugin McDonald et al.[25]. redbiom is a service for downloading data from Qiita Gonzalez et al.[3]

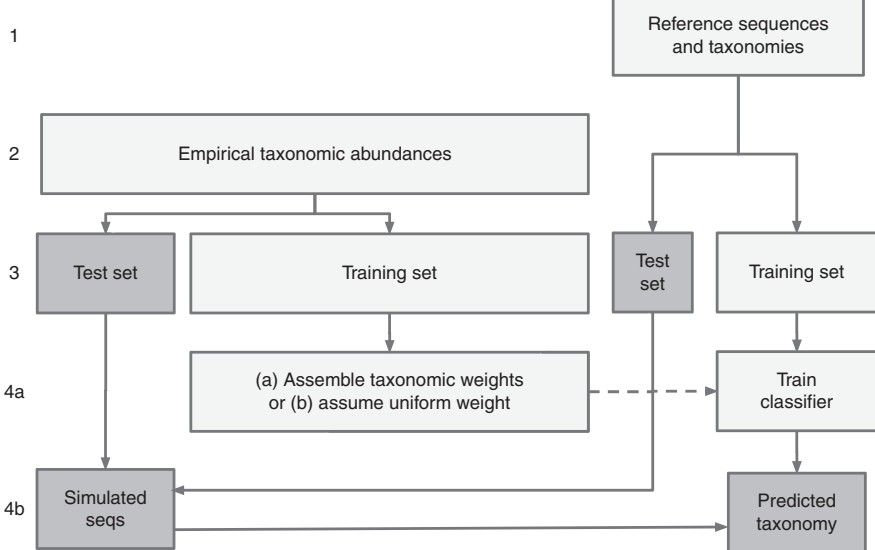

**Fig. 7** Cross validation workflow. Cross validation on the reference sequences ensured that a classifier was only ever asked to classify unseen sequences. Cross validation on the empirical samples used sequences from the test set of reference sequences to simulate samples with the same taxonomic abundances as the empirical samples, and ensured that bespoke and average weights were never derived from the samples on which they were tested

unobserved weight default) uniform offset (generate-class-weights) to form bespoke weights. The resulting weights are used to retrain the naive Bayes classifier to create a classifier under the bespoke weights assumption. In our experiments, which are detailed below, this procedure was modified slightly to accommodate cross validation and compilation of taxonomic weights from a variety of sources.

Internally, q2-feature-classifier uses the multinomial naive Bayes classifier provided by scikit-learn (see http://scikit-learn.org/stable/modules/naive_bayes.html). Loosely, the naive Bayes classifier finds the taxon that maximises the expression $P(T|S) = P(S|T) \times P(T)/P(S)$, where $P(T)$ and $P(S)$ are the probabilities of observing a taxon $T$ and sequence $S$ respectively, and $P(S|T)$ and $P(T|S)$ are the conditional probabilities of observing a sequence $S$ given a taxon $T$ and a taxon $T$ given a species $S$, respectively. The probabilities are estimated under questionable assumptions (the term naive refers specifically to the way $P(S|T)$ is calculated). The goal is not to provide a realistic model of reality; the goal is to predict taxa given

sequences. When taxonomic weights are provided, they are used directly as estimates of $P(T)$. For uniform weights, it is assumed that $P(T) = 1$. We note, however, that q2-feature-classifier is able to take taxonomic weights inputs for a variety of machine learning classifiers that are available in scikit-learn.

**Cross validation using empirical taxonomic abundance**. As mentioned above, to test classification accuracy using varying taxonomic weights, we developed a cross-validation strategy that accounted for the observed abundances of taxa in any given habitat.

Cross validation was used to analyse the effectiveness of setting the taxonomic weights for the q2-feature-classifier naive Bayes taxonomic classifier. A single cross-validation test follows the pattern (shown in Fig. 7, several steps are described in more detail below):

1. Obtain a set of reference sequences and reference taxonomies.
2. Obtain a set of samples for a given EMPO 3 habitat type, where each sample contains the number of reads observed for each taxon.
3. Perform stratified k-fold cross validation simultaneously on reference sequences and samples.
4. For each fold:

   a. Train a classifier on the training reference sequences, optionally incorporating read counts from the training samples to calculate taxonomic weights.
   b. Simulate samples that closely match the taxonomic abundances in the test samples using the test reference sequences, then classify them using the above classifier.

Step 2. Data were obtained as detailed above. Taxonomic abundances were estimated using the naive Bayes classifier under uniform weights using balanced settings[4], where the classifier was forced to classify to species level.

Step 3. We performed 5-fold cross validation in each instance. Standard stratification for 5-fold cross validation requires that at least five sequences exist for each taxonomy, which is not the case for the 99% identity Greengenes reference taxonomy. We, therefore, formed a stratum for each taxonomy for which five or more reference sequences existed (large taxonomies) and merged the remaining taxonomies (small taxonomies) into those strata. A single large taxonomy was chosen for each small taxonomy by training a naive Bayes classifier on the large taxonomies, classifying the reference sequences in the small taxonomies, then voting weighted by confidence. Shuffled stratified 5-fold cross validation was then implemented using a standard library call to scikit-learn[45].

Cross validation was performed simultaneously on samples and reference sequences. Sample cross validation was not stratified.

Step 4a. Each sample consisted of a set of taxonomies and their abundances. Taxonomic weights were formed by aggregating those counts across the training samples. As a result of the merged strata in Step 3, some taxonomies that were present in the bespoke weights were not present amongst the taxonomies of the training sequences. Any such taxonomy was mapped to the nearest taxonomy that was present amongst the taxonomies represented by the training sequences, as measured by the voting system from Step 3.

Step 4b. Samples were simulated by drawing sequences from the test sequences in such a way as to closely resemble the taxonomic abundances of the test samples. Again as a result of the merged strata in Step 3, some taxonomies that were present in the test samples were not present in the taxonomies of the test sequences. In the same way as for Step 4a, any missing species-level taxonomy was mapped to the closest taxonomy for a sequence present in the test sequences. Once missing taxonomies were resolved, samples were simulated by drawing test sequences as evenly as possible from each taxonomy so that any read count was a whole number.

For the q2-feature-classifier naive Bayes classifiers that were reported in this study, we used the recommended balanced parameters as recommended for uniform weights[4]. That is, we used a confidence level of 0.7 in all cases. In Bokulich et al.[4], a confidence level of 0.92 was recommended for bespoke weights tested on mock communities. We tested the classifiers at this level but in all cases the results were dominated by the less conservative confidence level of 0.7.

F-measure and Bray-Curtis[46] dissimilarity were calculated for each sample and taxonomic level using the q2-quality-control QIIME 2 plugin (https://github.com/qiime2/q2-quality-control). F-measure for each fold was aggregated across samples by weighting by the total read count for each sample. Bray-Curtis dissimilarity was averaged across samples without weighting, but samples with less than 1000 reads were filtered out.

Error rates, or the proportion of reads not correctly classified, were calculated as follows. A classification was called correct only if the expected classification exactly matched the observed classification to the required taxonomic level. That is, if the expected classification did not contain classification all the way to that level because that species was not present in the training set, then the classification was called correct only if it was truncated at exactly the right level. Correct classification rates were again calculated for each sample and aggregated across samples by weighting by the total read count for each sample. Aggregation across folds and EMPO 3 habitats was evenly weighted.

**Confusion Index**. The degree to which species can be successfully resolved is directly related to the dissimilarity of their sequences. We sought to establish a property of the reference data and taxonomic weights that were related to the classification accuracy across EMPO 3 habitats. For any pair of DNA sequences, the critical quantities are their sequence and taxonomic dissimilarities. Sequence dissimilarity is measured as the Bray-Curtis dissimilarity of k-mer counts. Taxonomic dissimilarity is the depth (from species level) of the most recent common ancestor, e.g. zero for the same species, one for species within the same genus and seven for an Archaean versus a Bacterium.

The Confusion Index is then the log of the product of the probability that the sequence dissimilarity for any pair of sequences is less than a threshold (we selected 0.25) and the expectation of the taxonomic distance given that the sequence dissimilarity is less than 0.25. The expectation was calculated under the assumption

that the two sequences were sampled independently with probability given by their bespoke weights. That is,

$$\text{CI} = \log \sum_{i=1}^{n} \sum_{j=1}^{n} d_t(i,j) I(d_s(i,j) < 0.25) w(i) w(j)$$

where CI is the confusion index, $d_s(i,j)$ is the sequence dissimilarity between the $i$ th and $j$ th sequences, $d_t(i,j)$ is the taxonomic dissimilarity between the $i$ th and $j$ th sequences, $w(i)$ is the weight of the $i$ th sequence, and $I(\bullet)$ is the indicator function.

The confusion index quantifies how often a pair of taxa have nearly identical sequences but different taxonomies for a given set of taxonomic weights. One advantage of this quantity is that it can be estimated statistically by taking a random sample of pairs of sequences. In this study we sampled $10^8$ pairs of sequences for each calculation.

**Comparison of taxonomic classification for shotgun and amplicon sequencing**. The effect of using taxonomic weights derived from taxonomic classification of shotgun sequencing reads was determined using 5-fold cross validation, where each classifier was trained using taxonomic weights aggregated across the samples in the training set, then tested on 16S samples from a test set. TDR[4] was computed using the q2-quality-control QIIME 2 plugin. TDR is the fraction of taxa that were discovered in the shotgun sequencing sample that were also found in the amplicon sample.

**Reporting summary**. Further information on research design is available in the Nature Research Reporting Summary linked to this article.

## Data availability
The Qiita data used in this study have been deposited at https://doi.org/10.5281/zenodo.2548899. The HMP and NCBI data used in this study have been deposited at https://doi.org/10.5281/zenodo.2549777. All other relevant data is available upon request.

## Code availability
q2-clawback is available at https://github.com/BenKaehler/q2-clawback/releases/tag/0.0.4. All other code developed for this study is available at https://github.com/BenKaehler/paycheck/releases/tag/0.0.4.

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

## Acknowledgements

QIIME 2 development was primarily funded by NSF Awards 1565100 to J.G.C. and 1565057 to R.K. This work was supported by an NHMRC project grant APP1085372, awarded to G.A.H., J.G.C., and R.K.

## Author contributions

Conceived, designed, and performed experiments: B.D.K. and N.A.B. Designed and wrote clawback software: B.D.K. and N.A.B. Wrote paper: B.D.K., N.A.B., J.G.C., and G.A.H. Developed supporting software (redbiom): D.M. and R.K. Provided critical review of paper and results: D.M., R.K., J.G.C., and G.A.H.

## Competing interests

The authors declare no competing interests.
