## [Peer Review File · Nature Communications]

Reviewers' comments:

Reviewer #1 (Remarks to the Author):

Kaehler and colleagues present a method for improving the accuracy of taxonomic placement of amplicon sequences using pre-defined environment-specific taxonomic weights. The authors rigorously demonstrate that their approach significantly improves over uniform weights when classifying low-level taxa. In addition, the authors demonstrate that the improvement is due in part to error from closely related taxa that inhabit distinct niches. The software for deriving weights is provided open source for use with tools within the QIIME2 analysis environment.

The authors innovate on an important topic within the field of amplicon-based microbial community profiling. The manuscript, while short, is easy to read and well-argued. The online methods and supplementary materials are similarly clear and well-assembled. My "major comments" below mostly reflect items I would like to see the authors discuss, rather than critical flaws with the paper/analysis.

MAJOR COMMENTS

=====

* (Abstract etc.) "Species-level resolution is attainable" feels like an overstatement? Species-level resolution was also attainable before the authors' work in cases where amplicons could distinguish between species. The authors improve the accuracy and extent to which OTUs can be assigned to known species, but it's not clear to me that they pass any fundamental threshold for species-level resolution in this work. If there's some objective standard here (e.g. >X% accuracy or within Y% accuracy of metagenomic species profiling) that would help to support the claim.

* How does this method avoid reinforcing previous taxonomic classification errors? For example, if a clade has been commonly misclassified in prior profiles of a given environment type, won't upweighting this clade serve to reinforce misclassification going forward?

* As a potential discussion topic: What practice would you recommend if a person is working with a novel environment type, or a novel perturbation of a known environment type? Fig. S4 (for example) suggests that using bespoke weights from the wrong environment was sometimes worse than using uniform weights.

* As a potential discussion topic: Would it be possible to use this method in an iterative way without specifying an environment? E.g. quantify and classify taxa by normal means within a dataset (with uniform weighting), then use the inferred weights to reclassify the taxa?

* It would be useful to know how the weighting information is being used within the classifier. My only takeaway here was that the authors' classifier COULD use weighting information, but that the focus was evaluating different types of potential weights.

MINOR COMMENTS

=====

* Please describe the test data at least somewhat in the main text.

* (Ln 56-61) I see what the authors are aiming for here, but starting with the comparison between species-level accuracy using bespoke weights vs. GENUS-level accuracy with uniform weights was not intuitive. (I had to re-read to catch "genus" as I was expecting both comparisons to be on species.)

The non-significant P-value is given to emphasize similarity, but this is not appropriate.

* (Ln 79) How were the different sets of weights compared?

* (Ln 91 etc.) I'm not sure what the authors mean by "sequence topology"? Are they referring to the topology of the tree for the sequences?

* (Fig. 1 etc.) As a general rule with bar plots the size of the colored portion should be proportional to the value it represents. Layered bars like these break that rule. I would either show the bars in-full side-by-side OR just show the accuracy values as points rather than bars.

* (Fig. 1 etc.) These could also be modified to emphasize the cases where one set of weights performs significantly better than another (or when the difference is NOT significant, if that is more common).

Reviewer #2 (Remarks to the Author):

In this manuscript, the authors describe a strategy for utilising prior information for the task of classifying short sequence reads of microbes taxonomically. Specifically, they suggest converting known average abundances of microbial species in the environments under study into prior probabilities or "taxonomic weights" to inform the training of a naïve Bayesian classifier (which uses k-mer frequencies to assign each read). The authors demonstrate that this approach results in improved classification when compared to a vanilla Bayesian classifier that assumes the prior probabilities to be uniform. A software package is presented that can assemble such taxonomic weights from a repository, for a habitat type of interest.

I am fairly skeptical of this manuscript, for the reasons outlined below, and I would suggest this to be of limited interest to the wide and diverse readership of Nature Communications.

MAJOR ISSUES

a) lack of novelty

The Bayesian classifier used here (q2-feature-classifier) is not novel – and neither is the approach to apply taxonomic weights to it. In fact, the authors' own previous work (Bokulich et al., Microbiome 6:90 2018) does exactly that. There, e.g. in Figure 1c, the authors show the performance of a "NB-Bespoke" approach, which is exactly what is described in the current manuscript. What is left in terms of novelty, then, is a set of scripts to run this more routinely and to assemble the environment-specific weights more easily.

b) conceptual appropriateness

The new approach seems to strongly outperform alternative methods, both in the current manuscript as well as in the previous paper mentioned above. I would argue that this needs to be interpreted very carefully. Of course, having available a wealth of information about what to expect from a given measurement (averaging over multiple prior studies), can be expected to allow the interpretation/estimation of that measurement with greater success. But does this really constitute "measuring" in a scientific sense? For routine, technical surveys, the use of prior knowledge can be useful – but for generating new, valid data points in a scientific experiment, potentially testing a new hypothesis, the use of prior knowledge can be quite dangerous. Essentially, it can give a false sense of confidence, and it may suppress novel or unusual patterns in the data – in a way that is very difficult to catch later on.

MINOR ISSUES

c) interpretation of reported error rates

Given that the reported error rate is calculated on the fraction of wrongly classified reads, averaged on a per-read basis, the approach taken to estimating classification performance is dominated by a few major taxa and is therefore not representative of the classification error of the vast majority of species. The authors should evaluate the performance of their method also in a way that is representative of the expected error on a per-species basis.

d) overruling sequence identity

The prior information used by the proposed method might lead to the misclassification of query sequences in cases where a sequence with high nucleotide identity (i.e. unambiguously assignable) is assigned to a wrong species because of the prior probability. K-mers alone might not diagnose this, but alignment-based methods would detect such edge cases. The authors should investigate to what extent this can happen, for instance by considering two sequences from closely related species (with a few nucleotides difference) and quantifying if and to what extent differences in prior between these species can induce misclassifications.

e) ground truth

It was a bit unclear from the manuscript what constituted the taxonomic 'ground truth' for assessing classifier performance. Was this exclusively based on the same short-read amplicon sequences, for example in Figures 1 or 2? If so, it could be fairly noisy itself, and perhaps quite biased.

f) error propagation

Any contaminants, errors, and biases (sampling or primer bias) present in previously analyzed data will be propagated to the results of the data being analyzed. On the long term, this could lead to the propagation/amplification of issues from previous data when using this approach.

Minor comments regarding the software:

g) The software installed correctly, and seems to work as described.

h) It seems, however, to be mainly aimed at users with previous knowledge of qiime2, rather than at pipeline-agnostic bioinformatics users.

i) A conversion is required of fasta and fastq files to qiime2 formats. This adds unnecessary steps to the data analysis and complicates the analysis by the introduction of superfluous file formats.

Reviewer #3 (Remarks to the Author):

This manuscript details an improvement upon the Naive bayes approach of 16S rRNA classification by using priors/weights informed by existing classified metagenomic samples. The authors convincingly demonstrate that no matter which way you "cut it," using more intelligent priors results in improved performance for the classification of 16S metagenomic samples. This result, while not surprising from a theoretical perspective, is especially important given that naive Bayesian approaches have been utilized for quite some time, with less attention payed to the choice of priors.

The authors pay careful attention to support their claims with a variety of experiments and assessment measures and also provide a tool allowing other researchers to take advantage of this approach. I do hope that this pipeline will be integrated into the QIIME framework as one of the

default options to q2-feature-classifier, so researchers can take advantage of these improvements without needing to manually retrain the classifier using q2-clawback.

I would suggest addressing just one issue to improve the reception of this paper by the metagenomics community: while it is clear that the authors trained the weights from 21,513 samples (line 51) it is less clear how they generated the data in figures 1 and 2. Was a subsample of these 21,513 samples used to assess the performance, or were novel samples used? The supplementary material makes this more clear (lines S53 and following), but it would be helpful in the main text to mention this rather than just very briefly mentioning cross validation in the caption of figure 1 of the main text (especially given how taxonomic weights are utilized in the cross validation design) . In particular, while the supplementary material makes it more clear, from the main text a reader may not be able to ascertain that performance improvement can still be had when presenting this "informed Bayes" approach with a sample that contains novel taxa.

Minor typos and suggestions:

line 33: "at genus level" - > "at the genus level" (and throughout, another style issue, but I typically see "at the genus level" or "at the species level" instead of "at genus level" or "at species level")

line 35: "including RDP Classifier" - > "including the RDP Classifier" (and throughout, the authors seemed to drop articles when referring to tool names. This is a style issue, but it reads more naturally to call a tool eg. "the RDP classifier" rather than just "RDP classifier").

Response to reviewers

We thank all the reviewers for their thoughtful comments. In response to their suggestions, we have made numerous changes to the manuscript.

Reviewer #1 (Remarks to the Author)

Kaehler and colleagues present a method for improving the accuracy of taxonomic placement of amplicon sequences using pre-defined environment-specific taxonomic weights. The authors rigorously demonstrate that their approach significantly improves over uniform weights when classifying low-level taxa. In addition, the authors demonstrate that the improvement is due in part to error from closely related taxa that inhabit distinct niches. The software for deriving weights is provided open source for use with tools within the QIIME2 analysis environment.

The authors innovate on an important topic within the field of amplicon-based microbial community profiling. The manuscript, while short, is easy to read and well-argued. The online methods and supplementary materials are similarly clear and well-assembled. *My "major comments" below mostly reflect items I would like to see the authors discuss, rather than critical flaws with the paper/analysis.*

We thank the reviewer for the kind comments and guidance on improvement to the manuscript.

MAJOR COMMENTS

1.1 Reviewer: * (Abstract etc.) "Species-level resolution is attainable" feels like an overstatement? Species-level resolution was also attainable before the authors' work in cases where amplicons could distinguish between species. The authors improve the accuracy and extent to which OTUs can be assigned to known species, but it's not clear to me that they pass any fundamental threshold for species-level resolution in this work. If there's some objective standard here (e.g. >X% accuracy or within Y% accuracy of metagenomic species profiling) that would help to support the claim.

1.1 AU: We agree, and thank this reviewer for correctly noting our accidental overstatement. An objective measure would be more appropriate and we have changed this sentence to read:

At the species level, overall average error rates dropped from 25% to 14%, which is favourably comparable to error rates that existing classifiers achieve at the genus level (16%). (line 23)

1.2 Reviewer: How does this method avoid reinforcing previous taxonomic classification errors? For example, if a clade has been commonly misclassified in prior profiles of a given

environment type, won't upweighting this clade serve to reinforce misclassification going forward?

1.2 AU: We concur that this is an important topic. We now extend our discussion of this caveat in the discussion to the following:

Efforts to curate reference databases and the continued contribution of researchers to online microbiome data repositories will help refine and extend the ability to apply appropriate bespoke weights for sequence classification in diverse sample types. We expect such efforts will lead to improvement in bespoke classification results, and allow generalization to other data types. In common with other methods, bespoke classification is not immune to errors that result from poorly curated reference data. However, as bespoke classification starts from the raw reference and read data when weights are derived, its use does not lead to misclassifications being propagated through history. Careful database curation remains critical to the continued improvement of classification accuracy. (lines 160-168)

1.3 Reviewer: As a potential discussion topic: What practice would you recommend if a person is working with a novel environment type, or a novel perturbation of a known environment type? Fig. S4 (for example) suggests that using bespoke weights from the wrong environment was sometimes worse than using uniform weights.

1.3 AU: While wrong weights can be worse than uniform, average weights always outperformed uniform taxonomic weights and may be used as an alternative for completely novel sample types when there are no existing samples. We offer this recommendation in the discussion:

For other natural sample types that lack sufficient characterization for bespoke weight assembly, average weights estimated from global microbial species distributions are superior to uniform weights. ... For highly unusual sample distributions, e.g., in synthetic populations, we recommend compiling custom bespoke weights from existing samples. (lines 151-158)

1.4 Reviewer: As a potential discussion topic: Would it be possible to use this method in an iterative way without specifying an environment? E.g. quantify and classify taxa by normal means within a dataset (with uniform weighting), then use the inferred weights to reclassify the taxa?

1.4 AU: We thank the reviewer for this a promising suggestion. It is similar in its approach to the empirical Bayes method (as distinct from more general Bayesian methodology). However, we were unable to identify a precedent for it in the machine learning literature. It falls under the challenging area of imbalanced classes and we felt we would not have a firm theoretical basis for inference. As such, significant and thorough empirical testing would be required and we are reluctant to speculate on it in the manuscript.

1.5 Reviewer: It would be useful to know how the weighting information is being used within the classifier. My only takeaway here was that the authors' classifier COULD use weighting information, but that the focus was evaluating different types of potential weights.

1.5 AU: We thank the reviewer for noting this useful clarification, and have added the following paragraph to the Methods section:

Internally, q2-feature-classifier uses the multinomial naive Bayes classifier provided by scikit-learn (see http://scikit-learn.org/stable/modules/naive_bayes.html). Loosely, the naive Bayes classifier finds the taxon that maximises the expression $P(T|S) = P(S|T) \times P(T) / P(S)$, where $P(T)$ and $P(S)$ are the probabilities of observing a taxon T and sequence S respectively, and $P(S|T)$ and $P(T|S)$ are the conditional probabilities of observing a sequence S given a taxon T and a taxon T given a species S , respectively. The probabilities are estimated under questionable assumptions (the term “naive” refers specifically to the way $P(S|T)$ is calculated). The goal is not to provide a realistic model of reality; the goal is to predict taxa given sequences. When taxonomic weights are provided, they are used directly as estimates of $P(T)$. For uniform weights, it is assumed that $P(T) = 1$. We note, however, that q2-feature-classifier is able to take taxonomic weights inputs for a variety of machine learning classifiers that are available in scikit-learn. (lines 237-248)

MINOR COMMENTS

1.6 Reviewer: Please describe the test data at least somewhat in the main text.

1.6 AU: We now briefly describe the test set and cross-validation procedure in the results (lines 64-71), and provide full detail in the methods section, which is now part of the main text (lines 185-343 of the revised manuscript).

1.7 Reviewer: (Ln 56-61) I see what the authors are aiming for here, but starting with the comparison between species-level accuracy using bespoke weights vs. GENUS-level accuracy with uniform weights was not intuitive. (I had to re-read to catch "genus" as I was expecting both comparisons to be on species.) The non-significant P-value is given to emphasize similarity, but this is not appropriate.

1.7 AU: That is a great point with the flow of the text. We have switched the order of paragraphs so that species-to-species comparison comes first. In addition, we have added emphasis to the genus-to-species comparison to help clarify (new text is below). Finally, we have removed the use of the non-significant p-values.

To demonstrate that bespoke classification achieves both greater accuracy and greater depth of taxonomic classification, we compared the error rates of bespoke classification at the species level to uniform classification at genus level. (lines 84-87)

1.9 Reviewer: (Ln 79) How were the different sets of weights compared?

1.9 AU: We apologize for the confusion here, we had originally included this in the methods section. To improve the readability, we have moved the following paragraph from the Method section to precede the empirical results:

To test classification accuracy using varying taxonomic weights, we developed a novel cross-validation strategy that accounted for the observed abundances of taxa in any given habitat. This strategy ensured that a classifier was never asked to classify a sequence that had occurred in its training set or generate taxonomic abundances that had directly contributed to its input taxonomic weights. To our knowledge, our cross-validation strategy is the first to incorporate information about taxonomic weights in assessing taxonomic classifier performance. This situation is known in machine learning as imbalanced learning⁴⁴. (lines 64-70)

1.10 Reviewer: (Ln 91 etc.) I'm not sure what the authors mean by "sequence topology"? Are they referring to the topology of the tree for the sequences?

1.10 AU: "sequence topology" has been changed to "sequence similarity" throughout.

1.11 Reviewer: (Fig. 1 etc.) As a general rule with bar plots the size of the colored portion should be proportional to the value it represents. Layered bars like these break that rule. I would either show the bars in-full side-by-side OR just show the accuracy values as points rather than bars.

1.11 AU: We thank the reviewer for remarking on this confusing aspect of the visualization. The bar plots have been replaced with box plots throughout.

1.12 Reviewer: (Fig. 1 etc.) These could also be modified to emphasize the cases where one set of weights performs significantly better than another (or when the difference is NOT significant, if that is more common).

1.12 AU: A pleasant caveat as a result of the reviewer suggestion in 1.11 is that the new box plots give a visual indication of how the spread of results overlap. In terms of significance, we felt that the significance of unanimous ranking between uniform, average, and bespoke weights was the most useful to emphasize.

Reviewer #2 (Remarks to the Author)

In this manuscript, the authors describe a strategy for utilising prior information for the task of classifying short sequence reads of microbes taxonomically. Specifically, they suggest converting known average abundances of microbial species in the environments under study into prior probabilities or “taxonomic weights” to inform the training of a naïve Bayesian classifier (which uses k-mer frequencies to assign each read). The authors demonstrate that this approach results in improved classification when compared to a vanilla Bayesian classifier that assumes the prior probabilities to be uniform. A software package is presented that can assemble such taxonomic weights from a repository, for a habitat type of interest.

I am fairly skeptical of this manuscript, for the reasons outlined below, and I would suggest this to be of limited interest to the wide and diverse readership of Nature Communications.

We thank the reviewer for their review. We hope the improvements made to the manuscript guided by peer review, and our responses to the concerns raised, have helped to increase the interest in this work. We felt that this work would be of interest to the general readership of Nature Communications as we are aware of at least one machine learning methods manuscript specifically for microbiome data (Menzel et al Nat Comm 2016), many other machine learning methods manuscripts (Kavvas et al Nat Comm 2018, Li et al Nat Comm 2019, Yao et al Nat Comm 2017, etc), and many microbiome manuscripts which rely on machine learning for taxonomic classification (Obregon-Tito et al Nat Comm 2015, Schnorr et al Nat Comm 2014, Youngblut et al Nat Comm 2019, Lurgi et al Nat Comm 2019, etc).

MAJOR ISSUES

2.1 Reviewer: lack of novelty

The Bayesian classifier used here (q2-feature-classifier) is not novel – and neither is the approach to apply taxonomic weights to it. In fact, the authors’ own previous work (Bokulich et al., *Microbiome* 6:90 2018) does exactly that. There, e.g. in Figure 1c, the authors show the performance of a “NB-Bespoke” approach, which is exactly what is described in the current manuscript.

2.1 AU: We respectfully disagree with the reviewers interpretation of the overlap between the previous (*Microbiome* 6:90 2018) and current work. While we employed the same term they refer to markedly different quantities. Specifically:

- 1. The current work defines “empirical bespoke” weights as estimates of *weights obtained from real-world samples*. We test them on different, out-of-sample data.**
- 2. The previous work employed “mock bespoke” weights that were *actual parameters (not estimated)*. The values for the mock-bespoke weights were**

explicitly defined as the mixing proportions of the mock communities. They were tested on exactly the same data to which they were applied.

To reiterate, in the current work “bespoke” means *inferring weights from real-world samples* then testing them on different, out-of-sample data. To the best of our knowledge, this is the first effort to implement this approach in the microbiome field.

The focus of our earlier work was on benchmarking existing popular techniques for taxonomic classification. Here we present a new approach that has brought novel discoveries:

1. We prove that prior information gained from existing data is practically guaranteed to improve discrimination of species in real, unseen biological communities.
2. We show in detail why taxonomic classification has always been hard and that there is sufficient similarity between samples from the same habitat to markedly reduce that difficulty.

In order to help improve clarity in the main text, we have added the following sentence (lines 43-45).

We also tested a developmental feature that showed that knowing the mixing proportions of mock communities improved taxonomic classification accuracy.

2.2 Reviewer: conceptual appropriateness

The new approach seems to strongly outperform alternative methods, both in the current manuscript as well as in the previous paper mentioned above. I would argue that this needs to be interpreted very carefully. Of course, having available a wealth of information about what to expect from a given measurement (averaging over multiple prior studies), can be expected to allow the interpretation/estimation of that measurement with greater success. But does this really constitute “measuring” in a scientific sense? For routine, technical surveys, the use of prior knowledge can be useful – but for generating new, valid data points in a scientific experiment, potentially testing a new hypothesis, the use of prior knowledge can be quite dangerous. Essentially, it can give a false sense of confidence, and it may suppress novel or unusual patterns in the data – in a way that is very difficult to catch later on.

2.2 AU: The reviewer raises several interesting points here.

The first is that using our techniques to increase taxonomic classification accuracy might not be “measuring” in a scientific sense. By using careful out-of-sample testing techniques, as detailed in the Methods, our results provide clear evidence that these new techniques genuinely increase classification accuracy specifically for unseen samples.

The second concerns the potential impact of using prior knowledge. As we show below, we have made a considerable effort to find risks associated with using habitat-specific weights. We identified a minor increase in risk of misclassification of

singleton reads in a normal sample (see response to 2.3). The following is now stated in the Discussion:

If the presence or absence of singletons for a typical sample is critical to experimental design, then we advocate using amplicon sequence variants^{22,42} rather than taxonomic classification. (lines 154-156)

Finally, through extensive testing with different weighting schemes (Figure S4), we have shown that it is actually difficult to do worse than the current prevailing assumption of uniform weights. On this basis, we argue that there is a clear negative impact on inference from continuing to use existing techniques rather than our new techniques. The flexibility of the tools we present, however, do provide a means for researchers to evaluate the robustness of classification of the different weighting schemes for cases relevant to their study.

MINOR ISSUES

2.3 Reviewer: interpretation of reported error rates

Given that the reported error rate is calculated on the fraction of wrongly classified reads, averaged on a per-read basis, the approach taken to estimating classification performance is dominated by a few major taxa and is therefore not representative of the classification error of the vast majority of species. The authors should evaluate the performance of their method also in a way that is representative of the expected error on a per-species basis.

2.3 AU: Thank you for this suggestion. We have provided accuracy results using the qualitative metrics, as shown in Figures S7 and S8. We also provide error rates for individual microbial species in Figure S9. We have added new sections to the end of the Results, the Discussion, and an extended description in the Supplementary Results. In summary we found that while qualitative metrics sometimes enjoyed the same increase in accuracy as their quantitative counterparts, sometimes the improvements were not as great. On average, however, they were not worse than existing methods. We also found that species that have very low abundance were sometimes classified with less accuracy. We therefore added the above recommendation to the Discussion.

2.4 Reviewer: overruling sequence identity

The prior information used by the proposed method might lead to the misclassification of query sequences in cases where a sequence with high nucleotide identity (i.e. unambiguously assignable) is assigned to a wrong species because of the prior probability. K-mers alone might not diagnose this, but alignment-based methods would detect such edge cases. The authors should investigate to what extent this can happen, for instance by considering two sequences from closely

related species (with a few nucleotides difference) and quantifying if and to what extent differences in prior between these species can induce misclassifications.

2.4 AU: This scenario is what the confusion index was designed to test. While k-mers are a blunt tool in comparison to sequence alignment, alignments that are almost identical must have almost all k-mers in common, so the thresholding on low diversity between k-mer counts will catch this case. As such, our confusion index results (Figure S7) show that the likelihood of misclassifying a sequence because it is almost identical to two sequences from different species largely determines classifier performance when using bespoke weights. This provides quantification of the extent to which real sample abundances and taxonomic weights affect classification accuracy.

2.5 Reviewer: ground truth

It was a bit unclear from the manuscript what constituted the taxonomic 'ground truth' for assessing classifier performance. Was this exclusively based on the same short-read amplicon sequences, for example in Figures 1 or 2? If so, it could be fairly noisy itself, and perhaps quite biased.

2.5 AU: We thank the reviewer for highlighting this, and we have expanded details in the Methods section (lines 249-313). Taxonomic ground truth for test reads was the taxonomic identities that were assigned to them in the reference data set.

2.6 error propagation

Any contaminants, errors, and biases (sampling or primer bias) present in previously analyzed data will be propagated to the results of the data being analyzed. On the long term, this could lead to the propagation/amplification of issues from previous data when using this approach.

2.6 AU: We agree and have discussed these issues in the discussion (lines 165-168), text shown above in this response). We do wish to note that issues of error propagation are present in general for classification given biases in references, and are not specific to this method.

2.7 Reviewer: Minor comments regarding the software:

- g) The software installed correctly, and seems to work as described.
- h) It seems, however, to be mainly aimed at users with previous knowledge of qiime2, rather than at pipeline-agnostic bioinformatics users.
- i) A conversion is required of fasta and fastq files to qiime2 formats. This adds unnecessary steps to the data analysis and complicates the analysis by the introduction of superfluous file formats.

2.7 AU: The QIIME 2 taxonomy classifier is the only classification method we are aware of that accepts taxonomic weight information as an input. Additionally, we provide a QIIME 2 plugin in this work because assembling appropriate taxonomic

weights involves many steps and database-specific considerations, which should ideally be persistently located within these files for reproducible data tracking. QIIME 2 provides this functionality because those files are not actually superfluous formats; they are just zip archives containing the data in standard formats as well as data provenance to track the analysis steps performed (QIIME 2 qza files can be opened with any zip utility, such as WinZip or 7-zip). QIIME 2 also provides multiple user interfaces, allowing users to interact with this method via a Python 3 API, galaxy, as well as a command-line interface. So we feel that the decision to implement this method in a QIIME 2 plugin provides greater accessibility to a broad range of users than creating a pipeline-agnostic python package.

Reviewer #3 (Remarks to the Author)

This manuscript details an improvement upon the Naive bayes approach of 16S rRNA classification by using priors/weights informed by existing classified metagenomic samples. The authors convincingly demonstrate that no matter which way you “cut it,” using more intelligent priors results in improved performance for the classification of 16S metagenomic samples. This result, while not surprising from a theoretical perspective, is especially important given that naive Bayesian approaches have been utilized for quite some time, with less attention paid to the choice of priors.

We thank the reviewer for the kind words. We agree that while the results are not necessarily surprising, facilitating the community in being able to take advantage of this prior knowledge is something that we feel is valuable.

3.1 Reviewer: The authors pay careful attention to support their claims with a variety of experiments and assessment measures and also provide a tool allowing other researchers to take advantage of this approach. I do hope that this pipeline will be integrated into the QIIME framework as one of the default options to q2-feature-classifier, so researchers can take advantage of these improvements without needing to manually retrain the classifier using q2-clawback.

3.1 AU: Thank you for this suggestion. We now provide a repository for sharing pre-assembled taxonomic weights (<https://github.com/BenKaehler/readytowear>), which currently includes taxonomic weights for the empo_3 sample types studied in this manuscript and other commonly studied sample types.

3.2 Reviewer: I would suggest addressing just one issue to improve the reception of this paper by the metagenomics community: while it is clear that the authors trained the weights from 21,513 samples (line 51) it is less clear how they generated the data in figures 1 and 2. Was a subsample of these 21,513 samples used to assess the performance, or were novel samples used? The supplementary material makes this more clear (lines S53 and following), but it would be helpful in the main text to mention this rather than just very briefly mentioning cross validation in the caption of figure 1 of the main text (especially given how taxonomic

weights are utilized in the cross validation design) . In particular, while the supplementary material makes it more clear, from the main text a reader may not be able to ascertain that performance improvement can still be had when presenting this “informed Bayes” approach with a sample that contains novel taxa.

3.2 AU: We thank the reviewer for this suggestion -- we have moved the entire Online Methods section into the main manuscript in the Methods section and selectively into the Results section. We hope that this clarifies that all of our results are based on out-of-sample testing, so directly translate to improvements that are expected for unseen samples.

3.5 Reviewer: Minor typos and suggestions:

line 33: “at genus level” - >“at the genus level” (and throughout, another style issue, but I typically see “at the genus level” or “at the species level” instead of “at genus level” or “at species level”)

line 35: “including RDP Classifier” - > “including the RDP Classifier” (and throughout, the authors seemed to drop articles when referring to tool names. This is a style issue, but it reads more naturally to call a tool eg. “the RDP classifier” rather than just “RDP classifier”).

3.5 AU: Thank you, we have added articles to all of these examples and throughout. There are some exceptions: we have not provided articles for software packages with one-word names (e.g., “The QIIME 2”) as they seem awkward.

Reviewers' comments:

Reviewer #1 (Remarks to the Author):

The authors have provided a thorough and thoughtful set of responses to my initial review. I endorse the manuscript for publication in its revised form.

Reviewer #2 (Remarks to the Author):

The authors have revised and clarified their manuscript, and provide a solid rebuttal to the three reviewers.

However, the manuscript still needs a more open discussion of the conceptual limitations of the approach, and of the dangers of using prior information. Both reviewers #1 and #2 raise issues relating to the use of prior information.

For example, reviewer #1 asks: "how does this method avoid reinforcing previous taxonomic classification errors?". The answer that the authors give is not satisfactory; they essentially say, "yes there are errors in curated reference data, but this type of error propagation is widespread". However, this is not the point - there are two types of error propagation to consider here: a) errors that propagate from the reference data (full-length 16S, properly annotated, and b) errors that propagate from the abundance assumptions that are taken as priors for the various environments. Point a) cannot be avoided, and yes it affects most approaches. However, point b) is specific to this approach: the abundance priors are learned from high-throughput datasets, the quality of which can be much less controlled and which can have any number of biases, artifacts or omissions. These technical shortcomings may then be propagated without being accounted for, and this should be clearly discussed in the manuscript.

In response to my question on whether the approach constitutes "measuring" in a scientific sense, the authors answer that they use careful "out-of-sample" testing techniques, and that this proves that they are indeed measuring. Again, I respectfully disagree: "out-of-sample", yes, but not "out-of-environment". The environment is the same in a given cross-validation fold, both in the training as well as the test set, and this injects a signal that simply is not in the data (that part of the signal is "inferred", not "measured"). To show and discuss the extent of this effect, perhaps the authors could create some artificial test samples that represent a random mix of environments? Then, the authors could assign these very samples taxonomically using classifiers that have been trained using the various different environments as priors - this should give differing responses for the very same samples, depending on which set of priors was used. How different are these responses, then? How reproducible are these differences? How large are these differences relative to robustness from resampling and/or random noise? This should be tested and discussed, to raise awareness with users as to what the results really mean and whether or not the approach is justified for a specific research question at hand. It would also drive home the point that one would get a different "result" for a given data set, simply depending on what one declares the environment type to be.

Reviewer #3 (Remarks to the Author):

The authors have addressed all the issues I had raised, and I am satisfied with the manuscript. The addition of pre-computed weights is appreciated and will help this method gain wider use in the microbiome community. Thanks to other reviewer comments, the figures have been significantly

improved and give a better indication of the improvement to be had when including bespoke weights. Furthermore, the newly included main text Figure 3 provides additional convincing evidence for the superiority of bespoke weights when some information about the sampling environment is known. This and other improvements further corroborate my initial assessment that the method presented provides strong evidence for its conclusions.

Additionally, I am convinced that this manuscript will be of interest to the readership of Nature Communications given the ubiquity of studies that utilize marker-gene DNA for metagenomic studies. Considering that most 16S rRNA studies inevitably utilize naïve (uniform) priors, and that the authors show that this practice is not justifiable, this method has the potential to significantly improve future metagenomic studies. The method and results contained in the manuscript are indeed novel: even though this method concerns a "classic" approach (naïve Bayesian classifier), the addition of informed priors (and a pipeline to generate them) has not to my knowledge been performed previously.

Response to reviewers

We thank all the reviewers for their thoughtful comments, and for reviewing the revised version of this manuscript. Reviewers 1 and 3 had only positive comments in this round of reviews, so in this response letter we focus on the comments of Reviewer 2. We have addressed these comments as detailed below. Author responses appear in boldface text and are preceded with "AU":

Reviewer 2 (Remarks to the Author)

1. The authors have revised and clarified their manuscript, and provide a solid rebuttal to the three reviewers.

However, the manuscript still needs a more open discussion of the conceptual limitations of the approach, and of the dangers of using prior information. Both reviewers #1 and #2 raise issues relating to the use of prior information.

For example, reviewer #1 asks: "how does this method avoid reinforcing previous taxonomic classification errors?". The answer that the authors give is not satisfactory; they essentially say, "yes there are errors in curated reference data, but this type of error propagation is widespread".

However, this is not the point - there are two types of error propagation to consider here: a) errors that propagate from the reference data (full-length 16S, properly annotated, and b) errors that propagate from the abundance assumptions that are taken as priors for the various environments. Point a) cannot be avoided, and yes it affects most approaches. However, point b) is specific to this approach: the abundance priors are learned from high-throughput datasets, the quality of which can be much less controlled and which can have any number of biases, artifacts or omissions. These technical shortcomings may then be propagated without being accounted for, and this should be clearly discussed in the manuscript.

AU: We thank Reviewer 2 for these suggestions. The point that compilation of taxonomic weights are affected by these biases is now addressed explicitly in the discussion:

The use of empirical species distributions also creates a potential source of error for bespoke classification... (lines 185-196)

2. In response to my question on whether the approach constitutes "measuring" in a scientific sense, the authors answer that they use careful "out-of-sample" testing techniques, and that this proves that they are indeed measuring. Again, I respectfully disagree: "out-of-sample", yes, but not "out-of-environment". The environment is the same in a given cross-validation fold, both in the training as well as the test set, and this injects a signal that simply is not in the data (that part of the signal is "inferred",

not "measured"). To show and discuss the extent of this effect, perhaps the authors could create some artificial test samples that represent a random mix of environments? Then, the authors could assign these very samples taxonomically using classifiers that have been trained using the various different environments as priors - this should give differing responses for the very same samples, depending on which set of priors was used. How different are these responses, then?

How reproducible are these differences? How large are these differences relative to robustness from resampling and/or random noise? This should be tested and discussed, to raise awareness with users as to what the results really mean and whether or not the approach is justified for a specific research question at hand. It would also drive home the point that one would get a different "result" for a given data set, simply depending on what one declares the environment type to be.

AU: We thank the reviewer for clarifying and stressing this point about “out-of-environment” classification . We performed this “out-of-environment” classification test, but referred to it as *cross-habitat* classification: see Figures 3 and 4 and lines 101-119 and 169-179 in the revised manuscript. In brief, we found that *choosing an incorrect environment type was typically still superior* to using the uniform weights. There remain important qualifiers, which we discuss in detail here and in the manuscript.

For the 14 environment types tested here, we created taxonomic weights and applied those for classification of the other 13 “wrong” environment types (e.g., classification of non-saline soil sequences with animal distal gut taxonomic weights). We also created taxonomic weights by equally mixing the 14 environment types to test “average” weights on all 14 environment types (while maintaining strict out-of-sample abundance sampling). This yielded several important findings:

1. Cross-habitat weights outperform uniform classification on average; in 117 out of 182 comparisons in our simulation, cross-habitat classification yielded more accurate prediction than uniform classification. *This deliberately incorrect choice is a worst-case scenario. In reality, we anticipate error in choosing environment type is likely to be less. Additionally, if a user is uncertain regarding environment choice they can specify average weights instead, which our results demonstrated were always superior to uniform.*
2. The error imposed by cross-habitat weight classification is directly related to the similarity between the target environment and the “wrong” environment from which these weights were assembled. So while environment-specific weights are always superior, assembling weights from similar environment types yields results that are nearly as good, indicating that bespoke classifiers are always preferable except for poorly-characterised sample types.

Regarding the reproducibility of these results, we show that a substantial amount of degradation in classification accuracy is explainable by the difference between the bespoke weights for a given EMPO 3 habitat and the cross-habitat weights used for classification (Pearson $r^2 = 0.57$, Figure 4). See lines 114-119. As to robustness,

characterisation accuracy of taxonomic weights for a given environment should increase with the number of samples used to derive them, but we found performance increases with as few as 122 samples. See lines 165-168.

We expand on these ideas extensively in the revised results and discussion, including the point that this reviewer raises, that “one would get a different ‘result’ for a given data set, simply depending on what one declares the environment type to be” in the paragraph that commences:

Our key finding is that taxonomic classification is sensitive to taxonomic weight assumptions... (lines 169-183)

REVIEWERS' COMMENTS:

Reviewer #2 (Remarks to the Author):

The authors have addressed my remaining concerns, by further clarifying the potential risks and biases.